# Aberration Theory of a Flat, Aplanatic Metalens Doublet and the Design of a Meta-Microscope Objective Lens

**DOI:** 10.3390/s23229273

**Published:** 2023-11-19

**Authors:** Woojun Han, Jinsoo Jeong, Jaisoon Kim, Sun-Je Kim

**Affiliations:** 1Department of Physics, Myongji University, Myongjiro 116, Namdong, Cheoin-gu, Yongin 17058, Republic of Korea; maxwell59@naver.com (W.H.); 1010jsk@naver.com (J.K.); 2Hologram Research Center, Korea Electronics Technology Institute, 8 Floor, 11, World Cup buk-ro 54-gil, Mapo-gu, Seoul 03924, Republic of Korea; j.jeong@keti.re.kr

**Keywords:** metalens, aberration, aplanatic lens, doublet, microscope objective lens

## Abstract

A theoretical approach for reducing multiple monochromatic aberrations using a flat metalens doublet is proposed and verified through ray tracing simulations. The theoretical relation between the Abbe sine condition and the generalized Snell’s law is revealed in the doublet system. Starting from the Abbe aplanat design, minimization conditions of astigmatism and field curvature are derived. Based on the theory, a metalens doublet is semi-analytically optimized as a compact, practical-level meta-microscope objective lens working for a target wavelength. The proposed approach also reveals how to reduce lateral chromatism for an additional wavelength. The design degree of freedom and fundamental limits of the system are both rigorously analyzed in theory and verified through ray tracing simulations. It is expected that the proposed method will provide unprecedented practical opportunities for the design of advanced compact microscopic imaging or sensing systems.

## 1. Introduction

Recently, with the rise of the metasurface optics (meta-optics) field, the dielectric metalens has arisen as the most important element of the meta-optics field as it provides new opportunities for ultracompact, multifunctional imaging systems and optical sensors [1,2,3,4,5,6,7]. However, most studies have focused on the suggestion of new ideas and potential metalens applications [4,5,6,7] rather than improving the performance limits toward the practical industry standard for refractive optics. In terms of image performance, there have been many studies on the compensation of metalens aberrations, both monochromatic and chromatic ones [8,9,10,11,12,13,14,15,16,17,18]. The hyperbolic phase profile at a design wavelength can perfectly eliminate a spherical aberration regardless of the numerical aperture (NA). However, it shows poor non-paraxial imaging performance owing to severe off-axis aberrations. Many studies about compensating longitudinal chromatic aberrations imply that a large chromatic dispersion of high-index dielectric meta-atoms is inherently harmful for the suppression of chromatic aberrations [6,7].

On the other hand, several breakthroughs in singlet or doublet metalenses have been proposed to deal with monochromatic aberrations. In a singlet scheme, multiple primary aberrations can be reduced over wide field angles only by introducing a quadratic phase profile rather than a hyperbolic one [8,9,10]. It was revealed that the quadratic phase method enables good performance for a low-NA design, and it can be also explained via the Fourier transform property of a paraxial lens in Fourier optics [8,9,10]. A few decades ago, D. A. Buralli and G. M. Morris proposed a similar idea of monochromatic imaging using an external front stop and a quadratic phase lens [19]. However, the application of these methods is restricted to wide-angle photographic lenses rather than high-resolution microscopy. For a greater design degree of freedom and a multi-objective design, metalens doublets combined with additional front stops have been successfully demonstrated for a miniature camera with a near-diffraction limited modulation transfer function and largely suppressed primary aberrations [11,12,13,14,15,16]. Since the two seminal papers on metalens doublets by A. Faraon’s and F. Capasso’s groups [11,12], numerous studies on parameter optimizations of the doublet profiles of nanoantenna arrays for various objectives such as achromatism have been successfully achieved [13,14,15,16].

A physical understanding of cascaded metalens systems is lacking despite their significance in potential complex imaging systems designed as hybrid refractive–meta-optics systems. Despite rapid advances in the numerical optimization of optical metalenses, the practical benefits of a profound theoretical understanding of aberration control cannot be neglected. The theoretical design of a meta-optic system can facilitate multi-objective optimization as well as better error management considering manufacturing and mass production compared to brute force optimization with many parameters. It is obvious that cascading additional metalenses to a doublet system would be harmful for optical efficiency, the operation bandwidth, and the manufacturing cost. On the other hand, a singlet provides limited design capability for reducing multiple aberrations. In this context, we conducted a rigorous study on a doublet platform which can not only provide a moderate design degree of freedom but may also have more advantages when hybridized with refractive lenses for high-performance imaging systems [17].

In this paper, the basic design theory of an Abbe aplanat for a flat metalens doublet is proposed and verified. Compared to previous papers suggesting numerical optimization examples of doublets [11,12,13,14,15,16], a rigorous theoretical study on ray aberrations is proposed, and a semi-analytical design example of a monochromatic microscope objective lens is suggested. The systematic design enables the compensation of multiple monochromatic aberrations simultaneously, without the help of ray tracing optimization simulations. Moreover, we investigate the achievable lateral achromatic performance of our aplanatic doublet method and the theoretical limit.

## 2. Theory of Aplanatic Metalens Doublet

In the field of classical lens design, Abbe found that in a spherical, aberration-free system, when a parallel ray incident along the optical axis has the same effective focal length regardless of its height, coma is also removed. This condition can be described as $\rho_{1}/\sin u_{2}^{\prime }=-f$, where $\rho_{1}$ is the height of the incident ray, $u_{2}^{\prime }$ is the converging angle at the image plane, and f is the effective focal length [20]. This is known as an Abbe aplanat, and it is required that the principal plane must be a spherical surface of a radius f for a very far object (Figure 1). F. Aieta et al. proposed the theory and design of a spherically curved metalens singlet which has a radius equal to its focal length [18], but the fact that it must be curved is an obstacle to its practical use. Chen et al. demonstrated a 1× magnification system with a special case in which the intersecting plane is flat [21]. However, this case of unity magnification is the only exception when the principal plane is flat.

In this context, for a more general and practical design, we propose a method using a flat metalens doublet separated by a substrate. It can be used to satisfy the aplanatic condition when the focal length of the system is determined. The phase profiles of two different metalenses are analytically derived via the geometrical ray tracing of the on-axis field in the meridional plane. For simplicity, we adopted a bi-facial doublet scheme (Figure 1) on a single quartz wafer substrate [22]. The metalenses were set to be separated by the substrate, and the first metalens was chosen to be the aperture stop. The proposed theory can also be extended to a cascaded system of two singlets on different substrates. To implement the proposed aplanatic doublet, the relationships between the two phase profiles of the surfaces should be determined. As illustrated in Figure 1, a ray incident on the first surface with a height ρ1 must intersect with the optical axis at the back focal point with an angle of *u*_2′_ = sin^−1^(*n*_1_*ρ*_1_/*n*_2′_*f*) after passing through the two surfaces. Consequently, the height *ρ*_2_ = *L*’tan*u*_2′_ at the second surface is dependent on both *ρ*_1_ and *L*’ (the back focal length). Given the distance between the two surfaces as *T*, the refractive angle *u*_1′_ of the first surface (the incident angle of the second surface) is determined by the height difference between the rays on the two surfaces:(1)$$u_{1}^{\prime }=\tan^{-1}(\frac{\rho_{2}-\rho_{1}}{T})=\tan^{-1}[\frac{L^{\prime }\tan\{\sin^{-1}(n_{1}\rho_{1}/n_{2}^{\prime }f)\}-\rho_{1}}{T}]$$

Throughout this paper, ray tracing notation and sign convention are adopted from the textbook (the second edition of *Lens Design Fundamentals*) by R. Kingslake [21]. Therefore, the refraction angle at *ρ*_1_ after the first surface can be written with respect to *u*_1_, and by applying this to the generalized Snell’s law, the phase gradients of each surface can be obtained as follows (See Appendix A for detailed derivation processes and the case of finite-finite conjugate imaging).
(2)$$\frac{\partial \phi_{1}(\rho_{1})}{\partial \rho_{1}}=kn_{1}^{\prime }\sin\left(\tan^{-1}u_{1}^{\prime }\right),$$
(3)$$\frac{\partial \phi_{2}\left\{\rho_{2}\left(\rho_{1}\right)\right\}}{\partial \rho_{1}}=k\left\{\frac{n_{1}\rho_{1}}{f}-n_{1}^{\prime }\sin\left(\tan^{-1}u_{1}^{\prime }\right)\right\}.$$

A flat metalens doublet consisting of the two correlated phase gradients according to the above equations completely satisfies the Abbe sine condition. Hence, the spherical and coma aberrations of all orders are removed at a design wavelength.

## 3. Optimization and Simulation of Doublet Metalenses

### 3.1. Compensation of Astigmatism and Field Curvature

The metalens doublet meeting the aplanatic condition suggested in the previous section has aberration-free performance for on-axis objects but still has other off-axis aberrations. In this chapter, we present a method to minimize image blur due to astigmatism and field curvature, which degrade image quality seriously. If astigmatism and field curvature are additionally reduced, the designed doublet can be used for monochromatic microscopy since it is an aberration-corrected objective lens with large magnification. In a doublet satisfying the aplanatic condition, it is possible to calculate the third-order tangential astigmatism by tracing off-axis rays at various angles. The refraction angle of an off-axis ray passing through the *r* point on the first surface can be determined using Equation (2). To calculate the refraction angle when passing through point s on the second surface (Figure 2), it is necessary to trace the on-axis ray passing through this point and calculate the phase gradient. Following this series of processes, it is possible to determine the height of the off-axis ray on the image surface.
(4)$$\begin{array}{l} h_{I}(r_{1})=r_{1}+T\tan\left[\sin^{-1}\left\{\frac{1}{n_{1}^{\prime }}\left[\frac{1}{k}\frac{\partial \phi_{1}}{\partial \rho_{1}}(r_{1})+n_{1}\sin\left\{\theta_{1}(r_{1})\right\}\right]\right\}\right]\\ +L^{\prime }\tan\left[\sin^{-1}\left\{\frac{1}{n_{2}^{\prime }}\left[\left\{\frac{1}{k}\left(\frac{\partial \phi_{1}}{\partial \rho_{1}}(r_{1})+\frac{\partial \phi_{2}}{\partial \rho_{1}}\left[s_{2}\left\{s_{1}(r_{1})\right\}\right]\right)+n_{1}\sin\theta_{1}(r_{1})\right\}\right]\right\}\right] \end{array}$$

The detailed derivation processes can be found in Appendix A. The height difference of the two marginal rays passing through the edges of the front stop (*p* and *q*), measured on the image surface, can be approximated to the magnitude of the tangential blur. In general, since tangential astigmatism is three times larger than sagittal [16], image quality could be improved even if only tangential blur is minimized. When the magnification or effective focal length of a complete system is determined, the variable values that optimize the performance of the marginal field can be found by numerically solving the equations.

Using the proposed design method, the authors designed an infinity-corrected meta-microscope objective lens. In most cases, for variable values in the ranges of f≤T≤2f and $0.5f\leq L^{\prime }\leq 1.5f$, a combination of phase gradients with minimum image blur is determined (Figure 3a). The optimized phase gradients when the thickness of substrate and back focal length are 15.470 and 8.681 mm, respectively, are fitted as less than an MSE (Mean Squared Error) of 2.7 × 10^−11^ in the function $\frac{\partial \phi }{\partial \rho }=\sum_{n=1}^{6}A_{2n-1}\rho^{2n-1}$, and the performance is verified with the Binary 2 surface, which adds phase to the ray according to the polynomial expansion $\Phi (\rho )=\sum_{n=1}^{m}A_{2n}\rho^{2n}$, provided by Zemax Opticstudio (Figure 3b,c). The resultant design layout is suggested as shown in Figure 3d. With an effective focal length of 10 mm, the system has diffraction limited performance and an RMS wavefront error of less than λ/30 for entire fields (Figure 3f,g). The field curvature is less than the depth of focus ($DoF=\lambda /NA^{2}$), and distortion is −0.2%. The plot in Figure 3h confirms that the optimized doublet satisfies the Abbe condition at a high level and Figure 4 shows the simulated point spread function intensity results.

The overall process of aplanatic doublet design is summarized in Figure 5. Compared to commercial 20× microscope objective lenses, which generally have diffraction-limited performance with NA 0.3–0.6 in an FoV of ± 0.20~0.65 mm, the meta-microscope objective lens has similar performance for the target monochromatic wavelength, as can be seen in Table 1. The proposed principles also work well for a finite–finite imaging system (See Appendix A).

### 3.2. Compensation of Lateral Achromatic Aberration

To the best of our knowledge, this is the first time a doublet phase gradient theory that satisfies both the aplanatic condition and lateral achromatism (the same paraxial magnification and effective focal length) at two different wavelengths has been proposed. In this case, for a basic study, we used the geometric phase method on both surfaces of the doublet. This means that the encoded phase profiles, for cross-polarized circular polarization, are common to different wavelengths as they are determined only by twice the rotation angle of the nanoantenna at a certain position, while the deflection angles are wavelength-dependent. Since effective focal lengths are calculated based on paraxial approximation, the design condition simultaneously meets the aplanatic condition for the primary wavelength, and the lateral achromatic condition can be derived using the paraxial approximation as follows:(5)$$\frac{\lambda_{s}}{\lambda_{p}}\left[\left\{\frac{n_{p}}{T}\left(1-\frac{L\prime }{f}\right)\right\}+\frac{f}{L^{\prime }}\left\{1-\frac{\lambda_{s}n_{p}}{\lambda_{p}n_{s}}\left(1-\frac{L^{\prime }}{f}\right)\right\}\left\{\frac{1}{f}-\frac{n_{p}}{T}\left(1-\frac{L^{\prime }}{f}\right)\right\}\right]-\frac{1}{f}=0$$

Here, *λs* is the secondary wavelength. *n_p_* and *n_s_* represent the refractive indices at the primary and secondary wavelengths of the substrate, respectively. The detailed derivation processes of Equation (5) can be found in Appendix A. The authors aim to design a lateral achromatic objective lens while maintaining the same specifications with the lens suggested in the previous section. Based on Equation (5), the condition between the substrate thickness and back focal length is derived to have the same effective focal length of 10 mm at two different wavelengths (0.532 and 0.460 μm), as shown in Figure 6a. In addition to this condition, the authors optimized the design to minimize off-axis blur. However, due to errors in the aplanatic condition ($Error=\left(f-\sin u_{\alpha 2}^{\prime }/r_{1}\right)/f\times 100(\%)$, where *u*_α2′_ is the slope angle of the secondary wavelength ray measured in the image plane) at the 0.460 μm wavelength, the secondary wavelength rays are not perfectly focused in the design attempt, as shown in Figure 6b–f. This error is intrinsic in this approach due to the paraxial approximation, as mentioned above. Even when the maximum aplanatic condition error is less than 0.3%, by reducing the NA of the system to 0.1, the perimeter (outer zone of the metalens) performance is deteriorated because the off-axis performance is not exactly optimized based on the method proposed in the Section 3.1. Thus, it is concluded that there is a design limit to perfectly satisfying the additional aplanatic condition for the secondary wavelength if the geometric phase method is used for the demonstration of the phase profiles of the two surfaces. Exploiting the propagation phase method or a judicious combination of the propagation and geometric phase methods would be helpful to improve the perimeter performance of the large-scale doublet system.

If the dimension of the doublet metalens is decreased, the achromatic performance can be increased to near-diffraction-limited performance at both wavelengths since the paraxial approximation fits better. The authors suggest a lateral achromatic doublet design (primary wavelength *λ_p_* = 0.532 and secondary *λ_s_* = 0.460 μm) for a system with a much shorter focal length and lens diameter. The design results are presented in Figure 7. The design process is the same as for the larger one suggested in the manuscript. The thickness (1 mm) of the substrate is determined first, and the back focal length is then set to be a variable. With a back focal length of 125 μm, the focal length at each wavelength is equal to 562 μm, and the system exhibits diffraction-limited focusing performance at the primary wavelength (Figure 7a–d). The paraxial back focal length at *λ_s_* is 185 μm, and the size of the image completely matches the primary wavelength, as can be seen in the image simulation results of Figure 7e,f using Zemax Opticstudio 18.1, but it is confirmed that there is little image blur and a slight decrease in the MTF at *λ_s_*. However, the overall imaging performance at the secondary wavelength is much better than the larger-scale design presented in Figure 7.

When it comes to the demonstration of real nanostructures of the metalens and its focusing efficiency, a resonant meta-atom structure for geometric phase modulation, such as the well-known cuboid-type TiO_2_ nanofin [23], can be engineered to achieve high transmittance based on local periodic approximation (which is well known from numerous previous studies) via electromagnetic simulations (See Appendix A for details). The optimized design proposed in Appendix A is a good candidate to be utilized simultaneously in the two different metalens profiles of the doublet. The local phase is modulated only by variations in the local rotation angle of nanofin meta-atoms with the same dimensions.

## 4. Conclusions

In this article, the theory of a flat, aplanatic metalens doublet is derived and extended to compensate additional ray aberrations. The optimized doublet can be used as a meta-microscope, providing near-practical imaging performance for a target wavelength. Moreover, the proposed achromatic design theory based on geometric phase encoding could be further improved via the introduction of more complex phase-encoding methods. We envision that the proposed work will provide a physical insight for optical system engineers working on meta-optic or hybrid refractive–meta-optic systems. It is also expected that the proposed method might be particularly fruitful for ultracompact microscopic imaging or sensing applications.

## Figures and Tables

**Figure 1 sensors-23-09273-f001:**
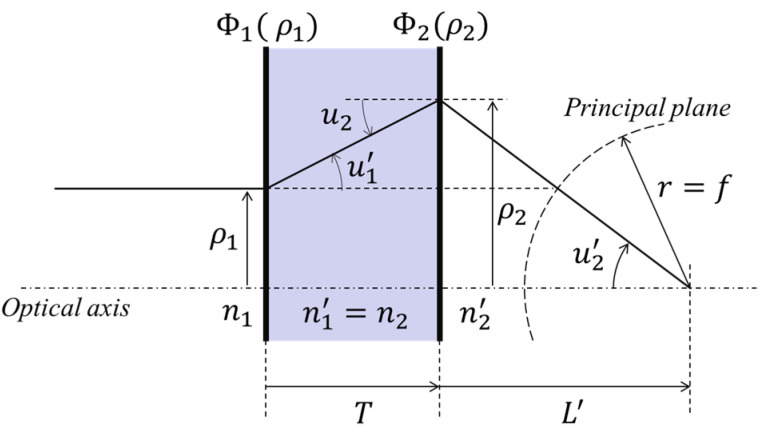
The Abbe aplanatic condition of a metalens doublet.

**Figure 2 sensors-23-09273-f002:**
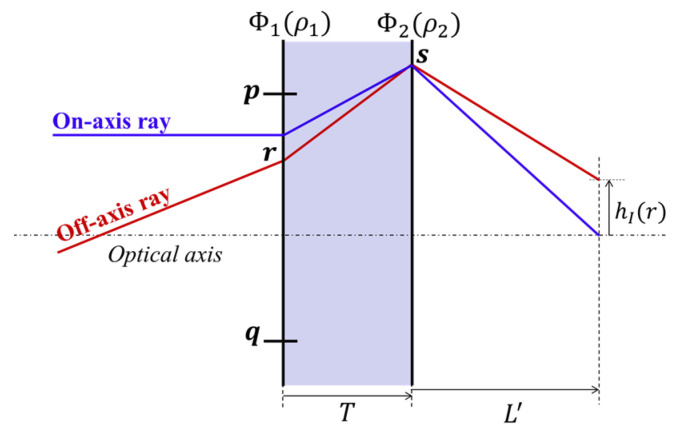
Scheme to investigate tangential image blur.

**Figure 3 sensors-23-09273-f003:**
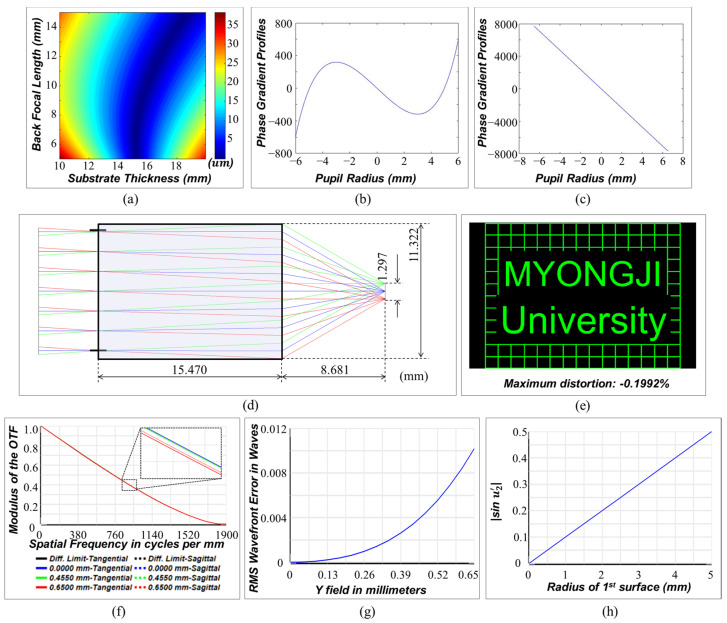
Aplanatic doublet design results. (**a**) The absolute value of the 3rd-order tangential astigmatism according to the variables *T* and *L* (**b**). (**c**) The phase gradient profiles of the 1st and 2nd surfaces of the optimized system. (**d**) The layout of the optimized aplanatic meta-microscope objective lens. (**e**) Image simulation result and distortion analysis. (**f**) MTF plot. The black, blue, green, and red lines represent the diffraction limit and the 0, 0.455 and 0.65 field positions (mm), respectively. The bold and dotted lines denote the tangential and sagittal planes, respectively. All field positions have achieved the diffraction limit. (**g**) RMS wavefront error vs. field plot. (**h**) The radius of the 1st surface and $\left|\sin u_{2}^{\prime }\right|$. $\left|\sin u_{2}^{\prime }\right|$ has a linear increase with respect to the radius of the 1st surface, and the slope is equal to *1/f*. This shows that the optimized system satisfies the aplanatic condition.

**Figure 4 sensors-23-09273-f004:**
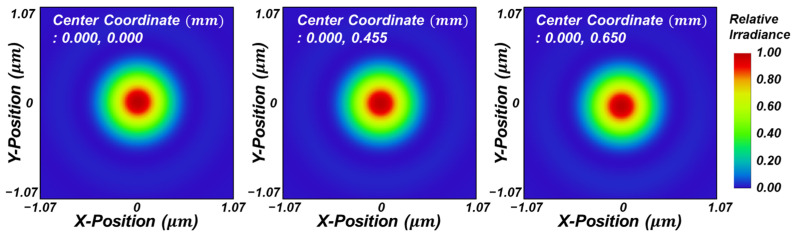
Simulated point spread function (PSF) intensity profiles of the aplanat doublet design result in the XY plane at a fixed focal plane.

**Figure 5 sensors-23-09273-f005:**
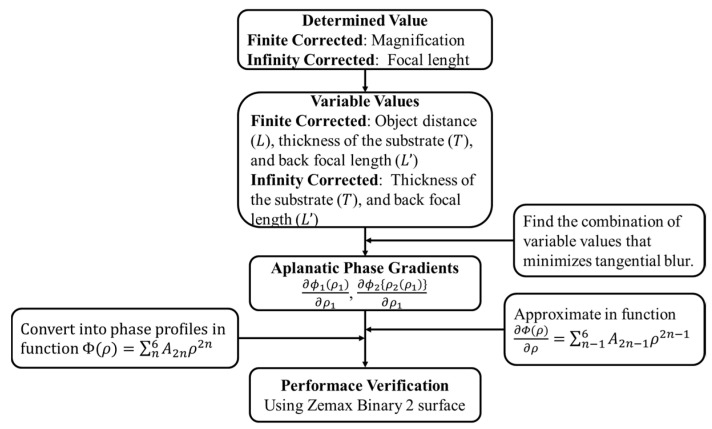
Flowchart of semi-analytic optimization processes for an aplanatic metalens doublet with minimal tangential blur.

**Figure 6 sensors-23-09273-f006:**
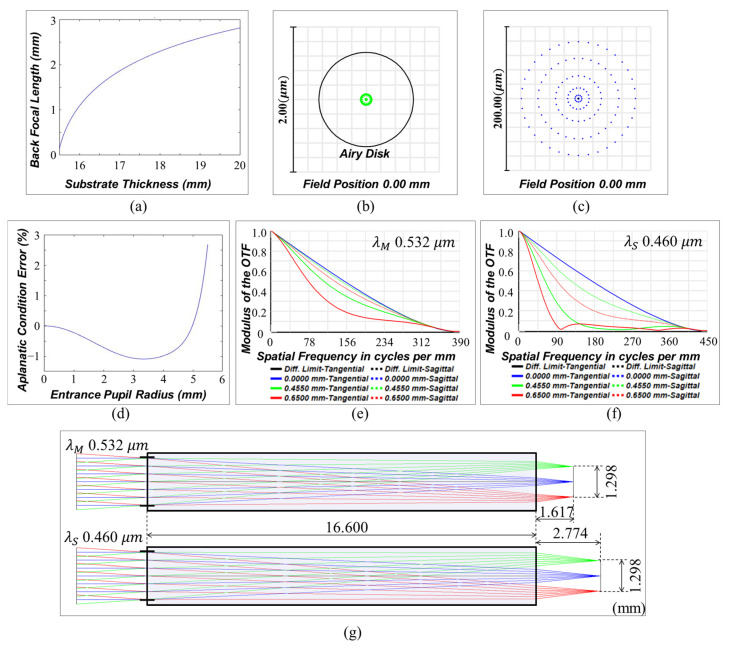
Lateral achromatic doublet design. (**a**) Lateral achromatic condition for wavelengths of 0.532 and 0.460 μm according to the thickness of the substrate and the back focal length when the effective focal length is 10 mm. (**b**,**c**) Spot diagrams on each image surface, 0.532 and 0.460 μm wavelengths, respectively. (**d**) Aplanatic condition error according to the pupil radius. MTF plots for (**e**) 0.532 and (**f**) 0.460 μm, respectively. The black, blue, green, and red lines represent the diffraction limit and the 0, 0.455, and 0.65 field positions (mm), respectively. The bold and dotted lines denote the tangential and sagittal planes, respectively. (**g**) Layout of a lateral achromatic objective lens with an NA of 0.1 and an effective focal length of 10 mm.

**Figure 7 sensors-23-09273-f007:**
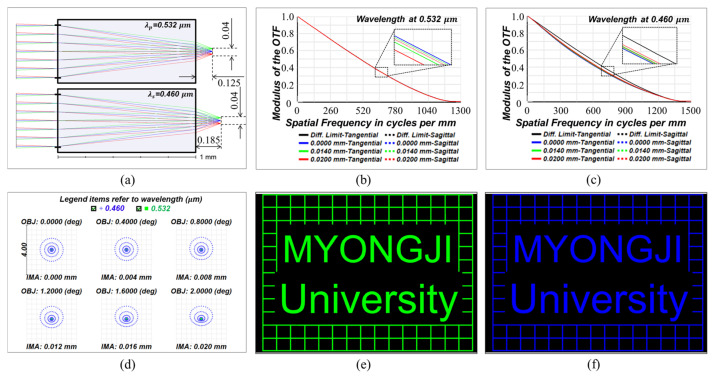
(**a**) Layout of a lateral achromatic doublet with a smaller size. MTF plots at wavelengths of (**b**) 0.532 and (**c**) 0.460 um, respectively. (**d**) Spot diagrams on the image surfaces of the two wavelengths. Blue and green dots refer to wavelengths of 0.460 and 0.532 μm, respectively. Image simulation results at the wavelengths of (**e**) 0.532 μm and (**f**) 0.460 μm, respectively. The size of each image is the same as 0.0303 W X 0.0196 H (mm).

**Table 1 sensors-23-09273-t001:** Comparison of a Mitutoyo M Plan Apo 20× (Mitutoyo, Kawasaki, Japan) and an aplanatic metamicroscope objective lens.

Specification	Commercial Microscope Objective Lens	Aplanatic Meta-Microscope Objective Lens
Magnification *	20×	20×
Focal length (mm)	10	10
NA	0.42	0.5
Resolving power (μm)	0.760	0.650
Diagonal FOV (mm)	1/2″ sensor format: ±0.2	±0.65
RMS wavefront error (mλ)	<0.06	<0.011
Distortion (%)	<|0.1|	−0.2
Wavelength (μm)	0.436 ~ 0.656	0.532
Working distance (mm)	20	8.681
Parfocal length (mm)	95	24.151

* The magnification is the value when using a tube lens with a focal length of 200 mm.

## Data Availability

The data presented in this study are available upon reasonable request from the corresponding author.

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
