# Peer review of "Aberration Theory of a Flat, Aplanatic Metalens Doublet and the Design of a Meta-Microscope Objective Lens"

_sensors, 2023, doi:10.3390/s23229273_

Round 1
Reviewer 1 Report
Comments and Suggestions for Authors
(Also please see the attachment) This paper provided the Aberration theory of aplanatic metalens doublet and a meta-microscope objective lens is designed. A Theoretical approach is adopted to reduce multiple monochromatic aberration. The relation of Abbe sine condition and the generalized Snell’s law is examined in the doublet system. The design of meta-microscope objective lens is just simulated in Zemax and it needs more analysis by FDTD method or similar to verify diffractive doublet metalens.
1. The notation of equation (ρ1/sinu2’=-f) in part 2 is different than figure 1. It’s better to use the same notation.
2. The MTF diagram of Figure 3f is plotted for a field, what about other field and also sagittal and tangential plots?
3. The sentence of “The optimized phase gradients are fitted as less than reduced chi square 1.2 E-12 in function when the object distance, thickness of substrate and back focal length are -223.462, 15.470 and 8.681 mm, respectively.” In S2.2 is not clear. Furthermore, the same issue in the sentence of “The optimized phase gradients when the thickness of substrate and back focal 152 length are 15.470 and 8.681 mm respectively are fitted as less than reduced chi square 1.2 153 E-12 in function” in paragraph, section 3.1. The expression of chi square 1.2 E-12 is not clear.
4. I suggest to add compare resolving power, magnification, resolution and contrast parameters to the table 1.
5. The compensation of lateral achromatic aberration is discussed in section 3.2 but there is no compensation of longitudinal achromatic aberration which is more important specially for the flat metalens. How do you solve the achromatic aberrations of doublet metalens?
6. The analysis in Figure 5 and 6 are simulated in Zemax. Do you think this analysis is enough to characterize the performance of meta-microscope objective lens? It may need to check doublet metalens design by FDTD method or similar methods.

Reviewer 2 Report
Comments and Suggestions for Authors
This work proposes the theoretical approach for reducing multiple monochromatic aberrations using flat doublet metalens via ray tracing simulation. The method and simulation analysis are demonstrated clearly in this manuscript. However, several previous works have published the same method of ray tracing simulation. Therefore, I recommend that the manuscript be accepted after revision. My comments are listed as follows:
1. What is the novel of the liquid-based metamaterial absorbers in this work, compared to the author's previous work?
2. The explanation of the statement “Abbe found that if ρ1/sinu2’=-f is satisfied in a spherical aberration-free system, coma is also removed” in line 75 should be described clearly. Please make sure the style of symbols are in the same format.
3. As we know, “doublet metalens" usually has a setup where two metalenses are used together. This means each metalens has a different phase, as presented in Figures 3(b) and 3(c). Could the author design the structure metasurface that can approach such phase gradient profile of both surfaces?
4. Addressing question 3, can such doublet metalens with the structure with such phase gradient profiles be fabricated?
5. The resolution of the image of figures in the manuscript is relatively low. The author should present high-resolution images for all figures in this manuscript to make the reader easily read.
6. To see the focusing performance of doublet metalens, it should be better to present the simulated intensity profiles in the xy plane at fixed position z.
7. To help the readers have a more comprehensive understanding of the new research on metamaterials, I suggest supplementing some of the latest works about metamaterials‐based photoelectric conversion: from microwave to optical range [Laser & Photonics Reviews 16, no. 3 (2022): 2100458], folding metamaterials with extremely strong electromagnetic resonance [Photonics Research 10, no. 9 (2022): 2215-2222], bricked subwavelength gratings: a tailorable on‐chip metamaterial topology [Laser & Photonics Reviews 15, no. 6 (2021): 2000478], terahertz liquid crystal programmable metasurface [Optics Letters 47, no. 7 (2022): 1891-1894], photonic [Laser & Photonics Reviews 16, no. 2 (2022): 2100328], and programmable controls to scattering properties of a radiation array [Laser & photonics reviews 15, no. 2 (2021): 2000449]

Round 2
Reviewer 2 Report
Comments and Suggestions for Authors
The author has great responses to the reviewer’s comments. I appreciated the authors’ effort in modifying their manuscript. The detail of However, there are two comments for the revised manuscript. Therefore, I would like to accept this manuscript.